# Keypoint-based morphological signature for large-scale neuroimage analysis

## Abstract

We present an image keypoint-based morphological signature that can be used to efficiently assess the pair-wise whole-brain similarity for large MRI datasets. Similarity is assessed via Jaccard-like measure of set overlap based on the proportion of keypoints shared by an image pair, which may be evaluated in $O(N \, log \, N)$ computational complexity given a set of $N$ images using nearest neighbor indexing. Image retrieval experiments combine four large public neuroimage datasets including the Human Connectome Project (HCP) (Van Essen et al., 2013), the Alzheimer's Disease Neuroimaging Initiative (ADNI) (Jack Jr et al., 2008) and the Open Access Series of Imaging Studies (OASIS) (Marcus et al., 2007), for a total of $N = 7536$ T1-weighted MRIs of 3334 unique subjects. Our method identifies all pairs of same-subjects images based on a simple threshold, and revealed a number of previously unknown subject labeling errors

**Keywords:** neuroimage analysis, neuromorphological signature, individual variability, salient image keypoints, MRI

## 1. Introduction

Increasingly large neuroimaging datasets and GPU-based machine learning algorithms offer the opportunity to study links between neuroanatomy and covariates such as genetic proximity. The primary challenge is the algorithmic complexity of this task, which is $O(N^2)$ for a set $N$ images and quickly becomes intractable for large datasets. A secondary challenge is data sparsity, there are typically very few data per subject (ex. one MRI), which makes it challenging to investigate the variability of individuals or family member labels via GPU-based machine learning algorithms. In this work, we present an efficient keypoint-based neuromorphological signature to investigate individual identification from multiple public well known large databases, as first described in (Chauvin et al., 2020). This approach, based on a 3D implementation of highly successful scale invariant feature transform (SIFT) (Lowe, 2004) from computer vision, has several advantages including robustness to global image transforms (avoiding the need of a computationally expensive image to image registration), occlusions (due to the local nature of keypoints), and efficient in identifying correspondances between generic intensity patterns in large image sets, due to highly efficient K-nearest neighbor (KNN) keypoint indexing algorithms operating $O(NlogN)$ complexity. The workflow used in this work is summarized in Figure 1.

## 2. Material & Methods

Data used in this experiment consist in T1w MR images from 4 datasets: HCP Q4, ADNI 1, OASIS 1 and OASIS 3, for a total of 8152 images. To focus our analysis on cortical and

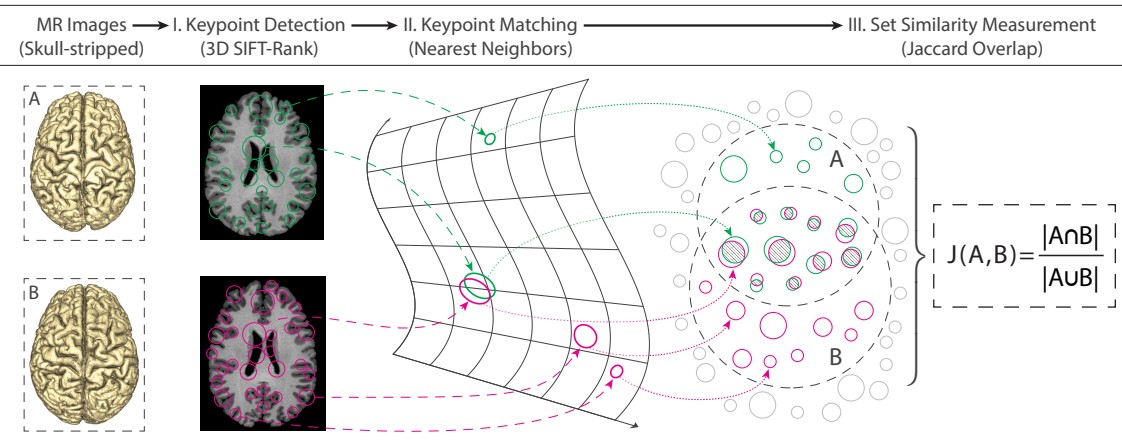

Figure 1: The workflow for computing the Jaccard overlap $J(A, B)$ similarity score between two images A and B. **Step I.** SIFT-Rank keypoints are extracted from skull-stripped MRI data. **Step II.** Similar keypoints are identified between images using a K-nearest neighbor search. **Step III.** The Jaccard overlap is computed as ratio of the intersection vs. the union of keypoint sets

subcortical structures, images were skullstripped using FreeSurfer v6.0. 616 images failed the pre-processing pipeline due to artefacts or low signal-to-noise ratio, resulting in a dataset of 7536 images of 3334 unique subjects. Each image is represented by a neuromorphological signature, a set of local keypoints $\bar{f}_i$, extracted and encoded using the 3D SIFT-Rank algorithm (Toews and Wells, 2013) with an average of 2130 keypoints/image. A GPU implementation of the algorithm in CUDA allows extraction in ¡ 2 seconds for typical 1mm resolution brain MRIs. Once keypoints are extracted, nearest neighbor indexing requires 0.35 seconds per image on an Intel Xeon Silver 4110@2.10Ghz via KD-tree lookup (Muja et al., 2009). Pairwise brain image similarity can be defined as the Jaccard overlap with soft set equivalence between keypoints in different images, or bag-of-features, based on the set intersection in equation (1). To account for the variable sampling density in the keypoint descriptor space, an adaptative kernel bandwidth $\alpha$ has been used for each keypoint descriptor $\bar{f}_i$, set to the squared distance to the closest nearest neighbor keypoint within the entire training set.

$$|A \cap B| = \sum_{\bar{f}_i \in A} \max_{\bar{f}_j \in NN_k(\bar{f}_i) \cap B} \exp\{-d^2(\bar{f}_i, \bar{f}_j)/2\alpha_i^2\}. \tag{1}$$

where $d(\bar{f}_i, \bar{f}_j)$ the Euclidean distance between descriptors $\bar{f}_i$ and $\bar{f}_j$, and $NN_k(\bar{f}_i)$ a set defined as the $K^{th}$ most similar descriptors to $f_i$ in the entire training set, identified with a k-nearest neighbor approach to reduce algorithmic complexity to $O(NlogN)$. To study the correlation between neuromorphological signatures and genetic proximity, each of the $N(N-1)/2 = 28,391,880$ pairs of signatures have been associated with a label according

to their possible relationship: same subject (SM), monozygotic twins (MZ), dizygotic twins (DZ), full-sibling (FS) and unrelated (UR), with labels of pairs across databases naively being assumed to be unrelated.

## 3. Results

As seen in Figure 2, the distribution of Jaccard-like distances, i.e. $d_T(A, B) = -log\ J(A, B)$, is highly unique for images of the same subjects (SM), with no overlap with other distributions. This particularity allowed us to classify images of the same subjects using their neuromorphological signatures with a simple threshold, revealing at the same time a number of outliers (see Figure 2). A thorough visual examination confirmed these outliers were mislabeled, and could be categorized into two groups: within database outliers and across databases outliers, containing respectively mislabeled images from the same database, and mislabeled images across databases, due to the naive assumption that subjects across databases were not related. Although it was expected to find shared subjects between OASIS 1 and 3 datasets, we also identified, more surprisingly, shared subjects between ADNI and OASIS databases. These results show that our neurmorphological signature succeeds

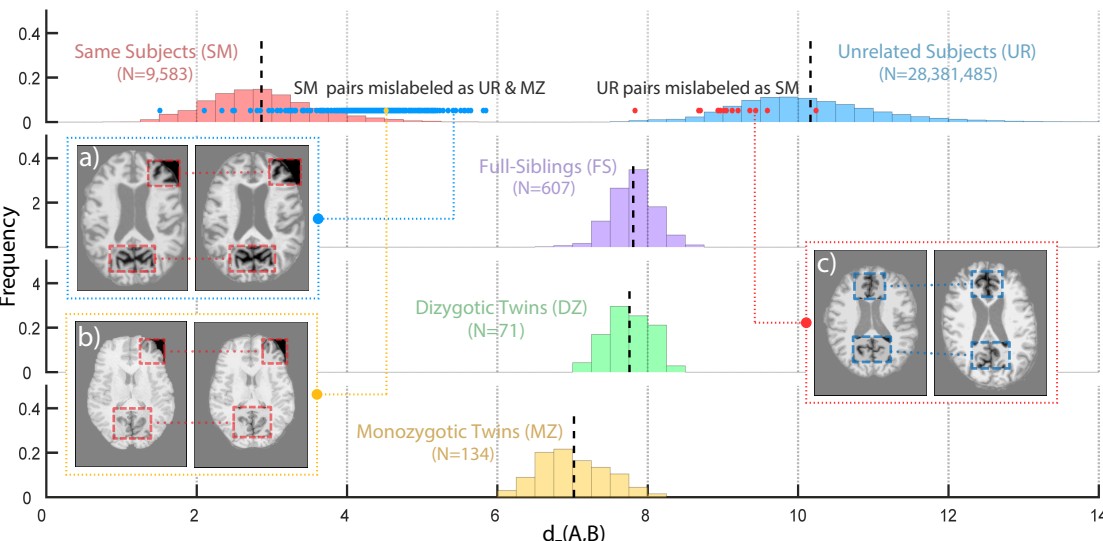

Figure 2: Distributions of the pairwise Jaccard-like distances $d_T(A, B)$ conditional on relationship labels. Dots indicate data labeling inconsistencies automatically flagged by unexpected Jaccard-like distance.

in capturing the individual variability in brain morphology on MR images, even with few samples per label. Its computational efficiency also allow our method to be used to study the impact of multiple factors, such as neurodegenerative diseases, aging, etc, on brain morphology across large populations, but also proved to be a powerful tool for curating large neuroimaging datasets.

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
