# OpenReview forum: "A Keypoint-based Morphological Signature for Large-scale Neuroimage Analysis"
_MIDL.io/2020/Conference — Submitted to MIDL 2020_

### Official Review · AnonReviewer2 · 2020-03-03
**Out of scope paper**

**Rating:** 3
**Confidence:** 3

**Review:**

Interesting short paper that proposes a keypoint-based morphological signature for large-scale
neuroimage analysis. Unfortunately it is not related (or the authors did not make any connection) with deep learning. This is the reason why the title of my review is "out of scope paper". It probably fits better on another conference.

---

### Official Review · AnonReviewer4 · 2020-03-10
**I reviewed the journal paper - excellent paper definitely worth presenting**

**Rating:** 4
**Confidence:** 5

**Review:**

I actually reviewed the journal version of this. It's an excellent paper which highlights the very important role that traditional feature based computer vision still has in medical imaging. Especially for tracking and comparing cortical features which deep learning is yet to prove it can do.

Minor comments:

I don't think the abstract does a good job of highlighting the key impact of the method. There is a lot of technical jargon. I would recommend re-summarising in plain english. In general the abstract is lacking a high level description of the motivations and the approach.

---

### Official Review · AnonReviewer1 · 2020-03-11
**Very impressive, but conventional machine learning**

**Rating:** 3
**Confidence:** 5

**Review:**

This paper presents a method to create a neuroimaging signature of an individual scan. The methods uses SIFT features and a kNN classifier. This paper validates the method based on 8152 scans. The authors analyse the correspondence between the signatures of all pairs of scans to analyse their correspondence. The results show the overlap measurement of the signatures is different between pairs of scans from the same individuals, from twins, from siblings and unrelated scans.

I find the analysis and its results very impressive and interesting. However, I am wondering if MIDL is the appropriate venue to present this, as no deep learning methodology is used.

---

### Official Review · AnonReviewer3 · 2020-03-13
**Interesting work but where is the deep learning?**

**Rating:** 2
**Confidence:** 3

**Review:**

The authors present a method to assess the similarity between pairs of images for large datasets that relies on the scale invariant feature transform. Both the method and the results described are interesting and of high quality. However, this work does not seem to be in line with the topic of the MIDL conference as no deep learning seems to be used in the analysis.

---

### Meta-Review · Area_Chair1 · 2020-04-06
**MetaReview of Paper294 by AreaChair1**

**Rating:** 1

**Metareview:**

This is a good paper, unfortunately it is out of the scope if MIDL since no deep learning is used in the proposed method.

**Paper Type:**

methodological development

---

### Decision · Program_Chairs · 2020-04-11

Reject